# Validation of a Korean Version of the Body-Appreciation Scale (K-BAS) in Young Women

**DOI:** 10.3390/ijerph19063426

**Published:** 2022-03-14

**Authors:** JungMin Lee, Shin-Jeong Kim, SoRa Kang

**Affiliations:** 1School of Nursing, Hallym University, Chuncheon 24252, Korea; j_lee0624@hallym.ac.kr; 2College of Nursing, Sungshin Women’s University, Seoul 02844, Korea; solarsolar81@naver.com

**Keywords:** body image, factor analysis, statistical, validation study, women

## Abstract

Despite concerns regarding body image in young Korean women, no measurement tool has yet been developed or is available. This study examined the validity and reliability of the Korean version of the Body-Appreciation Scale (K-BAS) to assess body image positivity among young women. For this purpose, convenience sampling was conducted using social network services. The participants were 245 women (N = 245) aged 20 to 40 years. Validity and reliability were examined using item analysis, factor analysis, and correlation with body mass index, body dissatisfaction, and Cronbach’s alpha. Twelve items were selected for the study. Two factors were extracted through factor analysis, explaining 64.82% of the variance and showing a good model fit in the K-BAS. The K-BAS score was negatively correlated with body mass index (*r* = −0.33, *p* < 0.001) and body dissatisfaction (*r* = −0.41, *p* < 0.001). Reliability was high, as indicated by a Cronbach’s alpha of 0.91. These results indicate that the K-BAS may serve as an appropriate instrument for measuring body image positivity among young Korean women. It may also be useful for identifying women with abnormal body perceptions.

## 1. Introduction

Body image is a complex and multidimensional construct reflecting an individual’s perceptions of and attitudes (i.e., feelings, thoughts, beliefs, and behaviors) toward their physical appearance [1,2]. Over the past decade, many studies have focused on the negative aspects of body image; however, positive body image has received substantial research attention in recent years [3,4,5,6]. A positive body image is not merely the opposite or absence of a negative body image but is a distinct construct with its own unique factors, including favorable opinions and acceptance of and respect for one’s body [4].

Body appreciation (BA) refers to one’s appreciation of the features, functionality, and health of one’s body and is a positive body image concept that has attracted the most attention [7]. It can be defined as “love and acceptance of one’s body and appreciation of its uniqueness and the functions it performs” [8]. Therefore, BA is considered a core aspect of a positive body image [9]. Researchers have found that BA is significantly associated with indices of well-being (e.g., optimism, self-esteem, proactive coping, and life satisfaction) and inversely associated with negative health outcomes (e.g., less depression, fewer unhealthy dieting behaviors) [7,9]. In addition, BA protects against body dissatisfaction [8]. These findings highlight the uniqueness of BA and demonstrate that its scope extends beyond mere satisfaction with the body or appearance [10].

Women tend to gain status and value through their appearance, whereas men tend to gain status from a wider range of qualities, such as intelligence, wealth, and power. Thus, women, especially when young, are placed under more pressure than men to have an ideal body shape; hence, they may be more critical and less appreciative of their bodies [3]. Women are often more strongly dissatisfied with their appearance because of sociocultural pressure based on the ideal of the female body being slender and thin. These stereotypes create intense pressure to conform and, consequently, influence young women’s self-esteem [11].

Asian culture and values include rigid social norms pertaining to the ideal body type as well as social comparisons in reference to those norms [5]. These affect Korean women, and social comparisons may thus influence their attitudes toward and satisfaction with their bodies. However, no study has investigated BA in the Korean population. Therefore, establishing a Body-Appreciation Scale (BAS) for young women who are most concerned with their bodies would help facilitate such body image studies and contribute to the prevention of body image-related issues, such as eating disorders.

To measure the BA construct, Avalos et al. [9] developed the BAS, a 13-item scale, using samples of adult women from the United States. Their main findings support the value of the BAS measure across a range of cultural contexts [8]. Research that validates body appreciation can inform positive psychology and may allow for improved intervention strategies that offer a more holistic approach for optimizing health and well-being [7,12]. Although many Koreans, especially young women, are greatly concerned about their body image, no measurement tools are available. Thus, finding ways to use existing tools through validation is preferable over developing new tools.

Despite concerns regarding young women’s body image, a scale measuring the level of concern has not yet been developed in Korea [13]. To measure body assessment in the West (i.e., Australia, Germany, and United States) as well as in other Eastern countries (i.e., in Indonesia Malaysia, and China), the BA instrument has frequently been cross-culturally adapted and used [14,15,16,17,18]. However, no Korean BA measuring tools with good psychometric properties are available. Therefore, this study sought to validate the K-BAS for use among young women concerned with their body image. The main research objectives were to (a) examine the validity and reliability of the K-BAS and (b) modify the scale and create a K-BAS that is more suitable for young Korean women.

## 2. Materials and Methods

### 2.1. Design

This methodological study examined the reliability and validity of the K-BAS based on the BA developed by Avalons et al. [9].

Before the study began, approval was granted by the H University Institutional Review Board of Korea (HIRB-2021-015). Potential participants were notified that (1) the survey would be anonymous, with the obtained data reported as group and not individual responses; (2) participants would have the right to withdraw from the study at any time without repercussions; (3) their privacy and confidentiality would be guaranteed.

### 2.2. Participants

The study sample comprised women aged between 20 and 40 years. According to Levinson’s model, the range of 20–40 years corresponds to early adulthood [19]. Age is limited because there may be differences in body image and other factors affecting body image according to the life cycle of adult women [20]. Jang and Yoo [21] stated that the body images of middle-aged adults in their 40s and 50s showed different characteristics than before, along with their preferred body images, standards, and ideal body images.

The inclusion criteria were as follows: all participants (1) were identified as female, (2) were aged 20 to 40 years, (3) were able to communicate in Korean, and (4) were able to access the Internet. Potential participants were excluded if they had psychological problems, such as depression, or had undergone plastic surgery, which may have affected their body image.

The number of samples was calculated based on the fact that more than 200 samples were required to obtain reliable factors for exploratory factor analysis [22]. With a 30% incomplete and drop-out rate anticipated, 260 participants were recruited, but the final sample size was 245 due to the incompletion rate.

### 2.3. Measurements

#### 2.3.1. BAS

Avalos et al. [9] developed the original BAS (13 items) to measure body appreciation. The items were designed to assess the extent to which women (a) held favorable opinions of their bodies; (b) accepted their bodies despite their weight, body shape, and imperfections; (c) respected their bodies by attending to their needs and engaging in healthy behaviors; and (d) protected their body image by rejecting unrealistic images of the thin, ideal prototype portrayed in the media. These aspects reflect the unconditional approval of and respect for the body as measured by the BA construct. The items are scored on a 5-point Likert scale (1 = never, 2 = seldom, 3 = sometimes, 4 = often, 5 = always), with higher scores reflecting greater BA. The Cronbach’s alpha in the original study was shown to have good internal consistency at 0.94. It was 0.91 in this study.

#### 2.3.2. Body Mass Index (BMI)

Participants self-reported their height in meters and weight in kilograms, and these were used to compute BMI using the formula “weight (kg)/(height) × (height)(m^2^)”. According to a meta-analysis, there was a statistically significant correlation between BA and BMI [4], and other studies have shown similar results [14,16,23,24].

#### 2.3.3. Body Dissatisfaction Scale (BDS)

Body dissatisfaction is defined as “a person’s negative thoughts about his or her own body” [25]. This measurement includes judgements about size and shape and generally involves a discrepancy between one’s body and an ideal body shape [26], typically based on photographic figures of computer-generated bodies. This scale can be used to measure body dissatisfaction by scoring bodies from 1 (leanest, emaciated) to 9 (obese) in ascending order of size, with each body scored as one body unit. The study participants were asked to choose the body they believed to be closest to their perceived body shape and the figure they most aspire to look like. The higher the BDS score between the perceived actual and ideal body, the greater the dissatisfaction.

### 2.4. Translation Procedures

To enable cross-cultural adaptation, the BAS was translated from English to Korean using a parallel back-translation procedure [27], supplemented by a committee review. In the first step, two bilingual independent translators, whose first language was Korean, translated the scale from English to Korean (T1, T2). Then, a third independent translator fluent in English and Korean performed a back-translation (T3). In the second step, both versions were reviewed by the first and second authors to identify and resolve inconsistencies between the original English and translated Korean versions. In the third step, two new independent translators naïve to the original BAS repeated the procedure (T4 and T5). Fourth, to consider the social and cultural aspects of the scale, the forward and back translations were re-examined by a committee consisting of the first, second, and third authors along with the translators (T4 and T5). It was agreed that no minor modifications were necessary to improve grammar clarity or item equivalence. The tool was finalized based on these discussions. Thirteen questionnaires were obtained from the K-BAS.

### 2.5. The K-BAS Validation Process

To create the K-BAS, the authors explained the study goals to a tool developer and received permission to verify its validity and reliability.

#### 2.5.1. Content Validity

Content validity was assessed by six experts (six nursing professors who majored in women’s health nursing) to determine the appropriateness of the forward-backward translated questionnaire. At this stage, pre-testing and cognitive interviewing were conducted by experts not only for clarity and flow but also for social and cultural characteristics. After content validity was verified, the final version of the K-BAS consisted of 13 items. Several words and phrases were revised to obtain a better fit with Korean culture.

#### 2.5.2. Construct Validity

##### Exploratory Factor Analysis (EFA)

EFA with varimax rotation was performed under oblique rotation. Data factorability was assessed using the Kaiser–Meyer–Olkin (KMO) measure of sampling adequacy, which should ideally be ≥0.80. Bartlett’s test of sphericity (which should be significant, *p* < 0.05) was also performed to evaluate factorability in terms of the magnitude of inter-correlations and sampling adequacy. The scree test of eigenvalues was plotted against these factors. Third, the proportion of variance accounted for by a factor was measured using the cumulative proportion (%) of the variance. Finally, we checked the pattern matrix, which contains the factor loading for each factor of primary interest.

##### Confirmatory Factor Analysis (EFA)

We performed a CFA on the two factors extracted using EFA. All items were loaded onto factors (>0.30 indicates fair loadings) [28]. Popular indices of model fit include the model chi-square, *df* and its *p*-value (which should not be significant), Steiger–Lind root mean square error of approximation and its 90% CI (with a value between 0.06 and 0.10 indicating adequate fit to the model), and Bentler comparative fit index (>0.95 indicates good model fit).

##### Criterion Validity

We compared the K-BAS with the BDS, which is widely used to measure body image.

##### Internal Consistency Reliability

An item-total correlation method was used to measure each questionnaire. Cronbach’s alpha was measured to estimate the internal consistency, and Guttman’s half-reliability coefficient was obtained using the half-section method.

### 2.6. Procedures

Convenience sampling, also known as “nonprobability sampling” [29], was used to recruit participants. We employed this sampling technique to recruit respondents using social networking services.

### 2.7. Data Analysis

Statistical analyses were performed using the SPSS statistics for Windows, version 25.0 (IBM Corp., Armonk, NY, USA) and AMOS statistics for Windows, version 23.0 (SPSS Inc., Chicago, IL, USA).

## 3. Results

### 3.1. General Characteristics of Participants

The study participants were 245 Korean women aged 20 to 40 (see Table 1). Their mean age was 30.40 ± 6.38 years. About 89.8% of the participants (n = 220) self-reported that they had experienced weight control in the past, and 45.0% (n = 99) were unsatisfied with the results of their weight control experience. A total of 57.6% of the participants (n = 141) were dissatisfied with the images of their bodies. The mean height of the participants was 1.63 ± 5.92 m, and their mean weight was 58.81 ± 10.21 kg. Their mean BMI was 22.03 ± 3.23 kg/m^2^.

### 3.2. Validity Test for K-BAS

#### 3.2.1. Content Validity

An evaluation by six experts found that the item-level content validity index (I-CVI) was in the range of 0.83 to 1.00, and the mean I-CVI was 0.96, which was higher than the cut-off value of 0.83 [30].

#### 3.2.2. Construct Validity 1: Item Analysis for Internal Consistency

The item-to-total correlations of the K-BAS were considered acceptable (see Table 2). While the “BA 11” item indicated a lower value than the standard at 0.30, there was no significant effect on internal consistency even when the item was removed. Thus, 12 items were selected.

#### 3.2.3. Construct Validity 2: EFA to Identify Hypothesized Components

An EFA with varimax rotation on the K-BAS items yielded two factors that explained the results and accounted for 64.81% of the variance. Prior to this, 13 items, such as the original tools, were put into an EFA; however, BA 11 was removed because it was not loaded in any of the derived factors. When the remaining 12 items were included in the EFA, Bartlett’s test of sphericity was found to be significant (χ^2^ = 1923.82, *df* = 66, *p* < 0.001), and the KMO measure was 0.93, which was suitable for factor analysis (see Table 3). Factors with an eigenvalue of one or more were extracted, and a marked decline in the slopes of the scree plot was observed. The final version of the K-BAS scale, consisting of a 12-item rotation factor loading, produced two factors with a loading range of 0.64 to 0.88 (see Table 3).

Two items, “BA 10: My feelings toward my body are positive, for the most part” and “BA13: Despite its imperfections, I still like my body”, were cross-loaded between the two factors. However, considering the content of the items, BA10 and BA13 were replaced with Factor 1 after in-depth discussion.

The first factor, containing eight items (BA1, BA2, BA3, BA4, BA5, BA6, BA7, BA10, and BA13), could be interpreted as a “body-esteem dimension” (eigenvalue, 5.89; variance explained, 49.05%). The second factor, consisting of four items (BA8, BA9, BA12, BA13), was related to the dimension of “body confidence” (eigenvalue, 1.89; variance explained, 15.77%; see Table 3).

#### 3.2.4. Construct 3: Appropriateness of CFA

The K-BAS model fit indices were χ^2^ = 133.85 (*df* = 53, *p* < 0.001), CMIN/DF = 2.53, RMSEA = 0.08 (90% CI: 0.06–0.10), TLI = 0.95, and CFI = 0.96 (see Table 3). Although χ^2^, which is sensitive to sample size, passed the bounds, all the other indices displayed the recommended level (see Table 4).

#### 3.2.5. Criterion Validity

The relationship between K-BAS and BD was examined. The K-BAS and BD scores were negatively correlated (*r* = −0.41, *p* < 0.001).

### 3.3. Reliability for K-BAS

The Cronbach’s α values were 0.91 for the 12 items on the K-BAS scale. The reliability values of each dimension were 0.94 for “body esteem” (Factor 1) and 0.63 for “body confidence” (Factor 2), which indicates high and moderate reliability [31]. Concerning the split-half reliability, Cronbach’s alpha was 0.76 and 0.91, and Guttmann’s half reliability coefficient was 0.87, indicating high reliability.

## 4. Discussion

This study aimed to validate a scale that can be used to measure BA in young Korean women. He et al. [5] reported that the BAS is the most widely used assessment measure for positive BAs. The BAS has been replicated in many Western countries. However, its use has not yet been examined in non-Western countries, especially East Asian countries, and there is no appropriate tool for measuring positive body image in Korea.

### 4.1. Factors Affecting BA

Our results indicate that BA can be reliably and validly measured and has two dimensions: body esteem and body confidence. The results demonstrated a positive correlation between BA and self-esteem in women. This result is consistent with the findings of Bhatti [3], who examined BA among Black women in the United Kingdom. The results suggest that positive body image increases satisfaction with one’s physical appearance and makes people more likely to feel comfortable and happy with the way they look [3]. Tatangelo et al. [32] also reported that self-esteem is critical for the development of a positive body image and the promotion of general well-being. This study suggests that a positive BA could be important because a high level of body image optimization can be beneficial throughout one’s lifetime.

A thin body is perhaps the most important current standard of attractiveness for young women [33]. Young women tend to internalize beauty ideals presented in the mass media, which are exceedingly difficult to achieve. As a result, even women with normal weight may experience negative emotions, feel ashamed of their bodies, and try to lose weight [34].

Previous studies have shown that BA is inversely associated with body dissatisfaction and eating disorders [4,12,35]. The present study produced similar results. Although most of the participants were within the normal BMI range, nine of the ten had experienced weight control, two out of five were unsatisfied with the results of their weight control experience, and more than half were dissatisfied with their bodies. People who identify as overweight despite being of normal weight or underweight and who control their weight recklessly should be given more attention because they may be at risk of health problems.

Overall, the results of this study show that positive body esteem and confidence can be considered as predictors and outcomes of positive body image. This makes it important to understand the factors affecting BA, which will provide basic data for the development of programs for the formation of a correct body image and mental health improvement among young women.

### 4.2. Validity

Our findings suggest that K-BAS scores have adequate content validity, strong construct validity, high internal consistency, and robust criterion validity. First, the content validity index (CVI) was found to be 0.96, suggesting 96% content agreement on all items of the K-BAS. Experts involved in the BAS assessed the changes in the items, deemed them appropriate considering the social and cultural aspects, and explained that the items were consistent with the content evaluated by the BAS. The terms chosen by the experts were changed slightly to suit the target population better. The sample displayed adequate content validity, which is in line with previous research suggesting that the factor structure of the BAS, a widely used measure of positive body image, can be adapted and utilized cross-culturally [5,24,36].

Second, although item-to-total correlations of the K-BAS were found to be suitable, item 11 had low factor loadings. Thus, it was deleted to improve the model fit. This decision differs from that of Swami and Jaafar [15], who decided to retain item 11 despite low factor loading. Swami and Chamorro-Premuzic [14] removed items 7 and 11 in their study on Malaysian women because their fit to the model was poor. Thus, the methods may differ depending on the BA construct used and the cultural dimensions of the setting. These results indicate that cross-cultural differences in the variables contribute to a positive BA.

The EFA results revealed that the construct validity of the K-BAS is adequate. The total cumulative percentage of the variance of the final two factors was 64.81%, and the scree plot and parallel analysis showed that the two significant factors may underlie the K-BAS. The two labeled factors were body esteem and body confidence. The first factor, body esteem, explained 49.05% of the variance and had an eigenvalue of 5.89 for nine items. The second factor, body confidence, covered three items, accounting for 15.77% of the variance, with an eigenvalue of 1.89. Indeed, in a previous study, both the KMO measure of sampling adequacy and the significance of Bartlett’s test of sphericity verified that the BAS items had adequate common variance for factor analysis [15,37,38]. They also verified that two factors were determined using a statistical criterion with an eigenvalue greater than one, with a steep decline between the rotated factors of the solution, similar to the results of our study [15,37,38].

The results of CFA to determine the factor structure of the K-BAS were adequate. Our results suggest that the two-factor structure satisfactorily explained the K-BAS: items 1–7, 10, and 13 as body esteem and items 8, 9, and 12 as body confidence. Although the nominal names of the factors are different, most previous studies have also shown that the BAS is divided into a two-factor structure toward the same items [14,24,37,38,39]. Taken together, the evidence suggests that the three items of the K-BAS (8, 9, and 12) are distinguishable from a one-dimensional factor. Based on the goodness-of-fit values, the results showed that the two-factor model, as suggested by a previous study involving adults from various countries, was the most robust and statistically significant anomaly in this study’s sample.

Finally, we verified the psychometric qualities of the K-BAS among women, satisfactory values of internal consistency, and a significant correlation between BMI and body dissatisfaction. The K-BAS had a significantly negative correlation with the previously validated BDS, which showed good criterion validity (*r* = −0.41, *p* < 0.001). Thus, the higher the BA, the lower the desire to lose weight. These findings mirror those of previous studies that a higher BA is linked to less weight and shape concerns [37,40,41,42].

Our results extend the work of Avalos et al. [9] in several ways. First, unlike the research of Avalos et al. [9], this study differs in that we ran EFA and produced two factors. This was done to reduce the number of variables to construct and explain the model with representative key elements to communicate information and knowledge more easily and effectively [43]. In addition, Avalos et al. [9] used all 13 items despite the relatively low loading of item 11 in the study (0.43 and 0.45, respectively); however, in this study, item 11 was removed because it did not belong to any factor. These results were similar in studies on populations in Indonesia, Malaysia, and China [14,15], indicating that intercultural differences in variables contribute to BA levels. Finally, we showed that body appreciation was negatively correlated with BD and BMI in women, while Avalos et al. [9] did not test the correlation between BD and BMI. The findings indicate that women’s body image is intricately linked to the degree to which they conform to social ideals about weight. Overall, both studies examined several key aspects of positive body image, and their psychometric properties were consistent.

Taken together, these results suggest that K-BAS scores have good psychometric properties for Korean women. The psychometric properties of the K-BAS meet many of the desired statistical requirements for a novel scale, which supports its use in multidisciplinary research. In general, the K-BAS has proven to be an appropriate scale for measuring BA among Korean women.

### 4.3. Reliability

Our findings show that the K-BAS has concrete reliability. The internal consistency of the BAS, as measured by Cronbach’s alpha of 13 items, was as high as 0.95 in the original study. It was also high in our study, which showed a Cronbach’s alpha of 0.91 for 12 items. Our findings are similar to those of Ng et al.’s [24] study on university students, which had an acceptable Cronbach’s alpha of 0.70 for the internal consistency of the Chinese version of the BAS. Similar results were found in Iran [44] and in the United States [18] for participants of similar ages (Cronbach’s alpha = 0.90 and 0.94, respectively). However, several previous studies [18,24] examined both sexes, while our study examined only women. Therefore, further studies should test the reliability of the K-BAS in a broader population.

### 4.4. Contributions and Limitations

This study makes several important contributions to the existing literature. First, robust translational procedures were employed to validate K-BAS. The tools in the Korean version are now readily available for future research on BA and its predictors. Second, the study’s findings have added new insights into the BA construct related to body image among South Korean women. We introduced a new construct, BA, which considers South Korea’s social and cultural characteristics, thus offering a tool with the good psychometric properties required to effectively measure BA constructs in Korea. Our study contributes to the larger body of research by providing a new reliable and valid K-BAS. If the two-dimensional structure of the K-BAS can be replicated in different cultural samples, it will serve as an important tool for assessing differences and similarities in body appreciation across cultures. This will not only provide a clearer picture of the BA concept but also facilitate cross-cultural comparisons where factor equivalence has been established. Third, this study indicates that the body esteem and confidence components of BA should be included in future health education and intervention programs.

The main limitations of this study are its use of convenience sampling and its relatively small sample size compared with previous studies on body image. Thus, to ensure that South Korea is represented as a whole, future research should use a much larger sample. Second, examining K-BSA in diverse groups could strengthen the reliability and validity of the tool. The applicability of our findings to other groups is limited because most of this study’s participants were young, middle-class women between 20 and 40 years of age. More groups should be surveyed before definitive conclusions about the BA of women are made. Third, longitudinal studies are required to identify other significant predictors of BA. Future research should identify other sociodemographic variables beyond country, age, and sex, which may be sources of non-invariant results. Finally, there may be errors in measuring anthropometric data. The results may be different if other measuring equipment are used, so this should be considered when measuring the height and weight of participants in future studies.

## 5. Conclusions

This study attempted to validate the BAS and determine whether it could be modified to create a K-BAS. These findings suggest that the K-BAS has good validity and reliability. Thus, this study supports the use of the K-BAS in measuring BA among young Korean women and contributes to the examination of the BAS across diverse populations. These results provide important implications for practice, among which is that the K-BAS may be a useful tool for the measurement of BA.

## Figures and Tables

**Table 1 ijerph-19-03426-t001:** Characteristics of participants (N = 245).

Characteristic	Category	n	(%)	Mean ± SD
Age (year)	20–24	64	26.1	
	25–29	57	23.3	
	30–34	58	23.7	
	35–40	66	26.9	
	Total			30.40 ± 6.38
Experience of weight control	Have	220	89.8	
	None	25	10.2	
Satisfaction with weight control (n = 220)	Very satisfied	16	7.3	
	Satisfied	85	38.6	
	Unsatisfied	99	45.0	
	Very unsatisfied	20	9.1	
Future weight-control plan	Have	202	82.4	
	None	43	17.6	
Satisfaction with body image	Very satisfied	7	2.9	
	Satisfied	64	26.1	
	Unsatisfied	141	57.6	
	Very unsatisfied	33	13.5	
Marital status	Married	116	47.3	
	Unmarried	126	51.4	
	Separated	3	1.2	
Educational level	High school	16	6.5	
	In college/university	46	18.8	
	college/university	162	66.1	
	Graduate school	18	7.3	
	etc.	3	1.2	
Occupation	Housewife	56	22.9	
	Student	49	20.0	
	Employed	140	57.1	
Household income	Under 1820	30	12.1	
(USD/monthly)	1820–under 2730	59	24.1	
	2730–under 3640	37	15.1	
	Above 3640	119	48.6	
Height (m)				1.63 ± 5.92
Weight (kg)				58.81 ± 10.21
BMI (kg/m^2^) †				
	Underweight (under 18.50)	25	10.2	
	Normal (18.51–24.99)	182	74.3	
	Obese (above 25.00)	38	15.5	
	total			22.03 ± 3.23

† Based on BMI for Asian populations (WHO Expert Consultation, 2004).

**Table 2 ijerph-19-03426-t002:** Internal consistency (corrected item-total correlation) (N = 245).

Item	Content	Item-Total Correlation	Alpha If Item Deleted
BA1	I respect my body	0.768	0.897
BA2	I feel good about my body	0.833	0.894
BA3	On the whole, I am satisfied with my body	0.784	0.896
BA4	Despite its flaws, I accept my body for what it is	0.703	0.900
BA5	I feel that my body has at least some good qualities	0.788	0.896
BA6	I take a positive attitude toward my body	0.828	0.894
BA7	I am attentive to my body’s needs	0.548	0.907
BA8	My self-worth is independent of my body shape or weight	0.389	0.916
BA9	I do not focus a lot of energy being concerned with my body shape or weight	0.379	0.915
BA10	My feelings toward my body are positive, for the most part	0.782	0.896
BA12	I do not allow unrealistically thin images of women presented in the media to affect my attitudes toward my body	0.301	0.919
BA13	Despite its imperfections, I still like my body	0.738	0.898

BA, Body Appreciation.

**Table 3 ijerph-19-03426-t003:** Result of exploratory factor analysis.

Item	Factor	
	1	2
BA 1	0.835	0.162
BA2	0.839	0.179
BA3	0.866	0.136
BA4	0.685	0.344
BA5	0.847	0.180
BA6	0.884	0.180
BA7	0.640	0.081
BA8	0.178	0.715
BA9	0.188	0.678
BA10	0.834	0.194
BA12	0.085	0.700
BA13	0.709	0.356
Eigenvalue	5.886	1.892
Total variance explained proportion (%)	49.050	15.768
Cumulative proportion (%)	49.050	64.818

BA, Body Appreciation.

**Table 4 ijerph-19-03426-t004:** Model’s goodness of fit.

Model	χ^2^	CMIN/DF	RMSEA	NFI	TLI	CFI
Criteria		<3	>0.05	>0.9	>0.9	>0.9
Results	133.85 (*df* = 53, *p* < 0.001)	2.53	0.079 (90% CI: 0.06–0.10)	0.932	0.947	0.957

CFI, comparative fit index; CI, confidence interval; CMIN/DF, chi-square minimum/degree of freedom; RMSEA, root mean square error of approximation; TLI, Tucker–Lewis index.

## Data Availability

The data presented in this study are available on request from the corresponding author. The data are not publicly available because it contains the participants’ personal information.

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
