# Peer review of "Validation of a Korean Version of the Body-Appreciation Scale (K-BAS) in Young Women"

_ijerph, 2022, doi:10.3390/ijerph19063426_

Round 1
Reviewer 1 Report
Validation of a Korean Version of the Body Appreciation Scale (K-BAS) in Young Women
Abstract
The objective is well identified and allows the future reader to know what, how and to whom the research has been carried out. However, it would be necessary to indicate why it is important to do this research.
It includes the main results and practical conclusions.
Introduction
Line 30-31. Please avoid truncating the syllables of a word. This is a spelling mistake (function-ality ... functio-nality).
Yes, we know that the template of the journal sets it up this way but authors have the possibility to avoid these spelling mistakes. Please check the whole text. Thank you.
The introduction is very well written and explains the topic of the study. However, it is necessary to indicate why this questionnaire should be validated for young women in South Korea. It should justify this need, and what studies support that BA is a concern among this population.
References to other similar studies on young women in other countries or circumstances should also be included. This would give much more value and credibility to the need for your research.
- Materials and Methods
2.1. Design
Ok, good explanation and good ethical behaviour before, during and after the research.
2.2. Participants
You should explain why you have chosen this sample (20-40 years of age) and not, for example, 18-50, 30-60, 25-50 ....
Self-concept of body image is linked to objective variables such as weight or BMI. Please justify in the article why these classificatory variables have been included in the participant questionnaire.
2.3. Measurements
2.3.1. BAS
Ok, good description and good justification
2.3.2. Body mass index (BMI)
Ok, good description and good justification
2.3.3. Body Dissatisfaction Scale (BDS)
Ok, good description and good justification
2.4. Translation Proceduret
Ok, good description and good justification
2.5. The K-BAS Validation Process
Ok, good description and good justification
2.5.1. Content Validity
Ok, good description and good justification
2.5.2. Construct Validity
This section faithfully follows the scientific procedure of construct validity.
- Results
3.1. General Characteristics of Participants
Lines 174-175. When you report "participants (n = 220) had experienced weight control"... How and when did you ask about weight control? This should be explained to future readers.
Why do you use BMI if you do not subsequently classify subjects as underweight, normal weight, overweight, obese? It makes no sense to have collected this information and then not provide information about whether there are differences in BDS... This must be justified.
...
Overall, this section is the most relevant of the entire article and the data provide new knowledge based on rigorous statistical science that will allow the applicability of this scale to the population of young women in South Korea. This is undoubtedly the best section of this article. Congratulations.
Author Response
Title: "Validation of a Korean Version of the Body Appreciation Scale (K-BAS) in Young Women ":
Manuscript ID: ijerph-1604492 Result: Minor Revision
We would like to express our appreciation for your extremely thoughtful suggestions. Your feedback was extremely helpful to strengthen our manuscript. As you will see below, we have been able to revise and improve the paper as a result of your valuable feedback.

Overall, we have made changes throughout the paper that address the points you have made as shown below. After correcting the manuscript according to the reviewers’ and editors’ comments, we got this paper revised by an academic revision company again. The corrected parts have been marked with Red Font and indicated with page numbers in the table below for easy reference.
Thank you again for taking the time to share your constructive feedback.
We also attach the file
(Table of Response to Reviewer)
Yours sincerely,
The authors

Reviewer 2 Report
This paper presents a new investigation on body appreciation in a sample of young Korean women. The study is based on responses reported by participants to the Korean version of the Body Appreciation Scale (K-BAS) during interviews using social network services. The topic covered is of scientific interest, the article is well written, and the statistical analysis is correct.
The discussion should be strengthened by more clearly indicating how this version differs from the original by Avalos et al (2005). In addition, although some of the study limitations are mentioned in the “Contributions and Limitations” paragraph, limitations related to the use of anthropometric data reported by participants should also be included and discussed.
Finally, here are my last minor remarks:
- page 3, line 134: You state that the assessment of content validity was conducted by six experts. However, it is not clear how these were chosen.
-page 4, lines 158-159: as criterion validity you claim to have "compared" the K-BAS to the BMI. How is it possible to compare two variables that are so different from each other? Was it also considered that the BMI was obtained from variables not directly measured?
-page 4, lines 177-178: taking into account the nature of anthropometric data - in addition, stature was referred to in m - reporting the averages (the stature in cm) with two decimals does not make sense. I suggest reducing them to one.
-Table 1: Consistent with what I have observed above, I suggest reducing the number of decimals of the anthropometric variables here as well.
-Table 3: I suggest eliminating this table because the results are already reported in the text (page 6 lines 201-203).
-Table 5: a final line indicating the end of the table is missing.
Author Response
Title: "Validation of a Korean Version of the Body Appreciation Scale (K-BAS) in Young Women ":
Manuscript ID: ijerph-1604492 Result: Minor Revision
We would like to express our appreciation for your extremely thoughtful suggestions. Your feedback was extremely helpful to strengthen our manuscript. As you will see below, we have been able to revise and improve the paper as a result of your valuable feedback.

Overall, we have made changes throughout the paper that address the points you have made as shown below. After correcting the manuscript according to the reviewers’ and editors’ comments, we got this paper revised by an academic revision company again. The corrected parts have been marked with Red Font and indicated with page numbers in the table below for easy reference.
Thank you again for taking the time to share your constructive feedback.
We also attach the file <Response to Reviewer>.
Yours sincerely,
The authors
